# The General Concept of the Revenue Model for Sustainability Growth

**Katarína Remeňová [1],\* , Jakub Kintler [2] and Nadežda Jankelová [1]**

1   Department of Management, University of Economics, 852 35 Bratislava, Slovakia;
    nadezda.jankelova@euba.sk
2   Department of Business Economy, University of Economics, 852 35 Bratislava, Slovakia;
    jakub.kintler@euba.sk
\*   Correspondence: katarina.remenova@euba.sk

**Abstract:** Since revenue streams provide financial stability for business development, companies are tasked with conducting an individual revenue model, that ensures their healthy and sustainable growth. Therefore, it is important to take into account the manner of creating revenues. This initiative relates to the revenue generation mechanism which describes how revenue is generated from the offer. Our research aims to find out whether the number of revenue streams (scope of the revenue model) and other financial and production indicators can predict the amount of revenue, as well as which factors can predict the scope of the revenue model. Concurrently we have focused on analyzing the sources of revenue streams and have selected key business variables in the wine producing industry. This sector had been selected as there is a significant difference between revenue streams based on manufacturing and those based on tourism activities. This concept was created by using theoretical knowledge about the value, pricing and segmentation. The goal of our research article was to design the General Concept of the Revenue Model and to identify variables with a strong influence on it. Results of the multiple linear regression analysis confirmed the significant influence of particular predictors on the scope of the revenue model, whereby the model explains up to 80% of the variance.

**Keywords:** revenue streams; revenue model; revenue streams of a winery

## 1. Introduction

Business models represent a platform for creating and delivering product value for customers [1]. Along with other components such as customer value proposition and relationships with partners, everything comes together to create the concept of a business model [2]. On the other hand, the revenue model refers to the platform for monetizing a customer's value [3]. Furthermore, the revenue model is incorporated into the business model and is also supported by the business model to monetize customer value. Therefore, we need to distinguish the revenue model from the business models, because of their different purposes.

Primarily, the revenue model theory provides useful information and a knowledge basis [4], largely from the on-line business environment [5,6].

In a business model, revenue should be attributed to different categories of channels. The Revenue model as an element of a business model measures the ability of a company to monetize values that are offered to the customers. This transformation generates incoming revenue streams. Therefore, it specifies how revenue streams are managed while individual components of the business model are involved in the generation of the revenue. It is based on the acquisition of knowledge about customer value, namely what kind of value is to be provided to the customer and which is the right channel to use for its distribution [7], as well as: how the value is to be communicated [8]; knowledge about

pricing; segmentation and correctly timing price optimization. According to authors, the revenue model explains how a company generates profit, while its economic character is perceived from the point of view of the revenue and cost logic [9–12].

A company's revenue model is composed of different revenue streams that can all have different pricing models [13], based on the offered customer value(s) [14,15]. The pricing mechanism provides an innovative approach to value creation from the product usefulness point of view of, perceived by the customer. Authors emphasize that the innovative revenue streams should be monitored, especially in those industries where production prevails. Efforts made in order to create products that influence the perception of such a product's value is governed by the revenue stream, through the product acceptance by customers [16]. Some researchers have aimed to predict value for consumers [17,18] and have obtained information about feelings and preferences based on physiological changes [19,20]. Customer value is created by key business activities [21–23] which can differ depending on the business model type [24,25]. It also allows for the company to adapt their business models to changes in market demands and a competitive environment [26], thereby better reflecting customers' needs [27]. Key business activities are required to: create and offer value proposition; reach the market; maintain customer relationships and earn revenues, and also illustrate how customer value is supported by different sets of activities [21].

Key business activities build the core of the business model. In the wine industry, it compares the production and sale of wine, sparkling wine, distillates, and non-alcoholic wine beverages. Additional business activities of wine-making companies include wine tasting, the organizing of social events for individuals and for legal entities, and the sale of accessory goods. The above described activities represent the way in which the company creates value for its customers. The system of revenue channels defines the form and how the value (product–service) is distributed to the customers [28]. Parameters analyzed within the system included the number of on-line revenue channels (own e-shop, partner on-line wine shops, partner e-shops), as well as the number of different types of off-line channels (distributors, wine shops, retail chains, HoReCa (Hotel/Restaurant/Catering).

According to MaRS start-up toolkit, it is necessary to be able to distinguish between four types of revenue models, determined by pricing decisions: the recurring revenue model; the transactional revenue model; project revenue model and services revenue model. Only the recurring revenue model appears to be a new trend in wine sales [29,30].

## 2. Materials and Methods

Our research aims to find out whether the number of revenue streams (the scope of the revenue model) and other financial and production indicators can predict the amount of revenue, as well as which factors can predict the scope of the revenue model. Concurrently, we have focused on analyzing sources of revenue streams and have compiled a general concept of the revenue model. We have looked at the revenue model from the perspective of the key business activities defined in the concept of business model Canvas [21,23,31] and other business model parameters such as the number of off-line channel types, the number of additional business activities and wine tourism activities [22].

The reason why we exactly focused on this theme was that there are insufficient scientific studies addressing the topic of revenue models in companies. As we have mentioned in the theoretical background, key business activities are the basic tool that companies use to create values that are subsequently served to the customers. In this process, it is important to set up the right price that reflects a customer's perception of the value that is linked with a concrete product or service. If company is able to set up a price that meets with the customer's perceived values, then it is able to ensure the long-term sustainability of its revenues and is also able to gain additional profit. We have focused our research according to this premise.

In the business management research field, there are many studies that deal with financial and production performance. Despite this, there are a lack of studies that are relevant to the performance of a particular revenue model. In the literature we can find lot of information about the price optimization,

but without a conceptually designed model of the revenue streams it does not make any sense. Because of this gap in the scientific literature, authors of this article designed the general model of the revenue streams. We can claim that existing research does not provide a clear decision-making process for creating a revenue model of a company.

The original research sample (n = 123) consists of Slovak wineries of all sizes. The wine-making industry belongs primarily to the manufacturing sector, however with an additional atypical revenue source, which is tourism. This specific condition led us to analyze the revenue model of wineries, which represents a novel field of research in the business environment.

Then, our survey was conducted using a questionnaire that focused on components of the business model, with open-ended and closed-ended questions that focused on revenue streams. A questionnaire was filled in by managers and owners of wineries.

*Validity and Reliability of the Research Tool*

Researchers have ensured the objectivity of measurements using data collected through the on-line form of the questionnaire and secondary sources, representing the financial statements and annual corporate reports. All variables represent interval or nominal data.

Individual consultations and structured interviews with individual wineries and researchers in the winery field led us to focus on three areas of issue outlined in the final version of the questionnaire. Thereafter, we proved the content validity and reliability of the research tool using Lawshe's content validity ratio (CVR) for measuring the validity [32]. Content validity was assessed using a panel of seven experts with professional expertise in the winery industry. Based on the Likert's scale, we collected and analyzed their viewpoints on the relevance of the 18 questionnaire items to measure the construct defined by these items to ensure the content validity of the instrument [33]. For the assessment, the following scale was used: „1 the item isn't representative of revenue model", „2 the item needs major revisions to be representative of the revenue model", „3 the item needs minor revisions to be representative of the revenue model", „4 the item is representative of the revenue model".

The CVR of each item didn´t decrease below the level of 0.7 and overall content validity index (CVI) reached the level of 0.95. Thereby we found this result to be representative.

The next step in the validation process of the questionnaire was to assess the reliability of our research tool [34]. Overall, the internal reliability measured by Cronbach's alpha reached the level of 0.71, which was considered as acceptable (see Table 1).

**Table 1.** Reliability Statistics with Cronbach's Alpha.

| Cronbach's Alpha | N of Items |
| --- | --- |
| 0.71 | 87 |

Source: own processing in PSPP.

The goal of our research article is to design the general revenue model. Concurrently, we have focused on analyzing sources of revenue streams and selected key business variables. This research goal was fulfilled with two steps. In the first step, it was necessary to design a general concept of the revenue model, which was created based on the theoretical knowledge about the value, pricing and segmentation (as we have presented in the Theoretical Background section). When creating the theoretical concept of the revenue model, we took into account the premise that company's revenue model can be composed of different revenue streams that can all have different pricing models [13].

The general Revenue Model could be widely used in various business models as a nested model. According to the theory of multilevel modelling, our nested model consists of the following three levels: TR: total revenue, RS: revenue stream and PM: pricing model. The first level represents the scope of revenues from all revenue streams of the company. Revenues are used as a variable. The second level is based on different revenue streams. This section identifies all business activities that should be monetized. The third level is created from pricing models. Each pricing model consists of a price,

a price metric, and a market segment. We emphasize that the appropriate price for each market segment needs to be adopted in the pricing model. Without this, it is not possible to perform a proper price optimization. At this level of the hierarchical model, the variables are price, metric and segment.

A Multiple Linear Regression analysis was applied to specify the dependence between the dependent variable revenue and predictors, such as the number of key business activities, the number of off-line channel types, the cultivated area, annual production, profit, years on the market and the number of revenue streams. The interrelationship of the variables is depicted as follows:

$$Y = \beta_0 + \beta_1 X_1 + \ldots \ldots \beta_j X_i + \varepsilon_i$$

where the $Y$ represents the dependent variable (as the expected value), the parameter $\beta_0$ represents the intercept, $\beta_1$ through $\beta_j$ are the slope coefficients, $X_1$ through $X_i$ are the independent variables and $\varepsilon_i$ represents the random error. The obtained data were analyzed in the PSPP (Perfect Statistics Professionally Presented) and EViews 11 statistical software.

## 3. Results and Discussion

Key activities of the business model of wine-producing companies consist of production and sale of wine. Some companies expanded their basic business model by incorporating additional business activities, such as the sale of accessory goods and wine tourism activities [35]. They include wine tasting, restaurant and accommodation services, and events. Key business activities define how the company creates values and satisfies customer needs with "value proposition".

Based on the descriptive statistical results we can say that the average revenue (in Euros) of wineries is $M = 79 \times 10^4$ with variability $SD = 22 \times 10^5$. The minimum value of the achieved revenue is $Min = -51{,}171$, maximum value is $Max = 17 \times 10^6$. The average gross margin value (%) is $M = 11.55$ ($SD = 127.26$) and gross margin interval is $Min = -917.60$, $Max = 329.07$. The average profit (in Euros) of wineries is $M = 13 \times 10^3$ ($SD = 34 \times 10^4$), the highest profit margin was $Max = EUR\ 25.10^5$. Based on the descriptive statistics of production indicators (Table 2) we can say that from the overall number of 100 cases, we do not have information about the annual production from 62 of them, and information about the cultivated area from 33 of them. The average annual wine production is $M = 26.38 \times 10^4$ litres, while the average cultivated area is $M = 50.70$ ha ($SD = 89.92$). The average number of years on the market of a wineries plant is $M = 13.75$ years. Lowest standard deviation of $SD = 12.64$ applies to the number of years on the market and the highest deviation of $SD = 66.4 \times 10^4$ to the annual production.

**Table 2.** Descriptive table for production indicators.

| Variable | N | Mean | Std. Dev | Variance | Mode | Median | Max | Min |
|----------|-----|-------------------|-------------------|---------------------|--------|--------|-------------------|------|
| Annual production | 38 | $26.38 \times 10^4$ | $66.4 \times 10^4$ | $44.8 \times 10^{10}$ | 22,500 | 42,500 | $37.5 \times 10^5$ | 3000 |
| Cultivated area | 67 | 50.70 | 89.92 | 8084.91 | 5 | 15 | 500.00 | 1.00 |
| On the market | 100 | 13.75 | 12.64 | 159.83 | - | 11 | 85.00 | 1.00 |

Source: own processing.

Based on descriptive statistics (Table 3), it is possible to state that highest variability between values reaches the variable Scope of Revenue Streams ($SD = 1.51$), while on the other hand the lowest variability between values is reported with No. of Key Business activities ($SD = 0.42$) with the average value of key activities $M = 1.11$. The average No. of revenue streams is $M = 2.65$, with the smallest number of streams from which wineries earn revenue being one and the highest number seven. The average No. of Additional Business Activities is $M = 1.54$ with $SD = 1.36$ together with the No. of Revenue Streams showing the highest dispersion of values from the mean.

**Table 3.** Descriptive table for business model elements.

| Variable | N | Mean | Std. Dev | Variance | Mode | Median | Max | Min |
|---|---|---|---|---|---|---|---|---|
| No. of On-line Channels | 100 | 0.71 | 0.62 | 0.39 | 1 | 1 | 3.00 | 0.00 |
| No. of Off-line channels types | 100 | 2.64 | 1.15 | 1.32 | 2 | 3 | 6.00 | 1.00 |
| Scope of Revenue Streams | 100 | 2.65 | 1.51 | 2.29 | 2 | 2 | 7.00 | 1.00 |
| No. of Additional Business activities | 100 | 1.54 | 1.36 | 1.85 | 1 | 1 | 6.00 | 0.00 |
| No. of Wine tourism activities | 100 | 1.34 | 1.24 | 1.54 | 1 | 1 | 5.00 | 0.00 |
| No. of Key Business activities | 100 | 1.11 | 0.42 | 0.18 | 1 | 1 | 3.00 | 1.00 |

Source: own processing.

The selection of the testing methods—Multiple linear regression analysis—was conditional on the fulfilment of the assumptions of the homoscedasticity, sphericity assumption, data bivariate correlation, autocorrelation of the residuals (using Dubrin–Watson test) [36] and the normality of data distribution.

Homogeneity tests were applied to verify the same response distribution, regarding the data from the questionnaire compared to the simulation sample. Then we can say that, the Kolmogorov–Smirnov test is valid, when a tested set of observations are from a completely specified continuous distribution [37]. As we can see in Table 4, the data distribution normality was tested through the Kolmogorov–Smirnov test. The assumption of homogeneity of variance also had to be tested using the Levene's test to ensure the equality of variances. The significance level of *p* for all variables is greater than 0.05, meaning that the test is not statistically significant and so the assumption of the normal distribution of the data can be confirmed. The results of the Levene's test for analyzing the sphericity and homogeneity of variance also does not confirm the violation of this assumption.

Then, we applied the Dubrin–Watson test to measure whether there is a correlation between the error term for one observation and the next. Our Dubrin–Watson statistic reached a level of 1.83, that indicates that there was no autocorrelation [38].

Since all tests did not confirm the violation of conditions for parametric testing, we can use the bivariate correlation and linear regression analysis. The results of the bivariate Pearson's correlation are referred in Table 5.

Examined variables such as No. of Additional Business activities, Gross Margin, No. of On-line Channels did not indicate the required statistically significant *p*-value level. Therefore, we need to redesign the model used.

To investigate our research questions concerning the revenue and the scope of the revenue model, we used a multiple linear regression analysis, where functional dependence was identified between the dependent variable Revenue and the independent variables such as the cultivated area, annual production, years on the market, number of key business activities, number of off-line channel types.

Characteristics of the non-financial and non-production variables are based on the Osterwalder's business model definition. Scope of revenue model is defined by the number of revenue streams, which depends on multiple independent variables (mentioned above).

As it turned out, the correlation between revenue and variables mentioned in Table 5, is for some variables statistically significant. Therefore, we have examined the predictive influence of individual variable on the size of the revenue.

Table 6 represents the aggregated results for the regression model summary and the ANOVA model for dependent variable revenue. In order to determine the relationship between the variables, Pearson's correlation coefficient was used. In regards to the variables, the results of the multiple linear regression analysis for the number of key business activities, number of off-line channel types, the cultivated area, annual production and years on the market, have confirmed their significant influence. The F-ratio shows whether the overall regression model is a good fit for the data. Outputs indicate that the independent variables statistically significantly predict the dependent variable, $F_{(6, 97)} = 186.46$, *p*-value = 0.000. The model explains up to 97% variance of the revenue.

Table 4. Results of Kolmogorov–Smirnov Test and Levene's statistics for cardinal variables.

| | Cultivated Area in Hectars | Annual Production | Revenue | Gross Margin | Profit | On the Market | No. of Core Business Activities | Wine Tourism Activities | No.of Additional Business Activities | No. of Revenue Streams | No. of Off-Line Channels Type | No. of On-Line Channels |
|---|---|---|---|---|---|---|---|---|---|---|---|---|
| N | 67 | 38 | 98 | 98 | 96 | 100 | 100 | 100 | 100 | 100 | 100 | 100 |
| Levene's statistics | 2.43 | 0.27 | 1.74 | 0.14 | 1.79 | 0.658 | 0.53 | 0.68 | 0.89 | 0.89 | 2.68 | 0.82 |
| Asymp. Sig. (2-tailed) | 0.074 | 0.763 | 0.101 | 0.939 | 0.06 | 0.078 | 0.140 | 0.568 | 0.451 | 0.451 | 0.055 | 0.488 |
| Kolmogorov- Smirnov Z | 0.38 | 1.14 | 0.52 | 3.29 | 3.42 | 1.59 | 0.68 | 2.68 | 2.54 | 2.56 | 2.11 | 3.09 |
| Asymp. Sig. (2-tailed) | 0.100 | 0.070 | 0.083 | 0.170 | 0.06 | 0.09 | 0.063 | 0.090 | 0.059 | 0.070 | 0.130 | 0.060 |

Source: own processing.

**Table 5.** Results of the bivariate correlation analysis.

| Factor | | Revenue | Revenue Streams |
|---|---|---|---|
| Cultivated area | Pearson Correlation | 0.66 | 0.29 |
| | Sig. (2-tailed) | 0.000 | 0.017 |
| Annual production | Pearson Correlation | 0.96 | 0.12 |
| | Sig. (2-tailed) | 0.000 | 0.464 |
| On the market | Pearson Correlation | 0.60 | 0.09 |
| | Sig. (2-tailed) | 0.000 | 0.360 |
| No. of Key Business activities | Pearson Correlation | 0.53 | 0.49 |
| | Sig. (2-tailed) | 0.000 | 0.000 |
| No. of Additional Business activities | Pearson Correlation | 0.04 | 0.96 |
| | Sig. (2-tailed) | 0.664 | 0.000 |
| Gross Margin | Pearson Correlation | 0.06 | 0.12 |
| | Sig. (2-tailed) | 0.563 | 0.233 |
| No. of Revenue Streams | Pearson Correlation | 0.19 | 1 |
| | Sig. (2-tailed) | 0.055 | 0.000 |
| No. of On-line Channels | Pearson Correlation | −0.06 | 0.07 |
| | Sig. (2-tailed) | 0.576 | 0.494 |
| No. of Off-line channels types | Pearson Correlation | 0.26 | 0.30 |
| | Sig. (2-tailed) | 0.010 | 0.002 |
| No. of Wine tourism activities | Pearson Correlation | 0.09 | 0.89 |
| | Sig. (2-tailed) | 0.380 | 0.000 |

Source: own processing.

**Table 6.** Linear Regression Model Summary & ANOVA results for dependent variable Revenue.

| Regression | R | $R^2$ | Mean Square | F | Sig. |
|---|---|---|---|---|---|
| Revenue | 0.98 | 0.97 | $20.28 \times 10^{12}$ | 186.46 | 0.000 |

Source: own processing.

In the next step, we looked over the statistical significance of each independent variable. As reported in Table 7, not all examined variables and the unstandardized coefficients are equal to zero in the sample- t $(\beta_{\text{Cultivated area}})$ = −2.64, *p*-value = 0.014; t $(\beta_{\text{No. of Key Business activities}})$ = 6.10, *p*-value = 0.000; t $(\beta_{\text{Annual production}})$ = 25.52, *p*-value = 0.000; t $(\beta_{\text{No. of Revenue Streams}})$ = 5.71, *p*-value = 0.000.

**Table 7.** Summary table of Coefficients for dependent variable Revenue.

| | $x_i$ | Unstandardized Coefficients | | | |
|---|---|---|---|---|---|
| | | B | Std. Error | t | Sig. |
| | (Constant) | −1,279,191.28 | 250,382.63 | −5.11 | 0.000 |
| | Cultivated area | −4516.51 | 1713.74 | −2.64 | 0.014 |
| | No. of key Business activities | 1,190,683.81 | 195,234.46 | 6.10 | 0.000 |
| | No. of Off-line channels type | 62,855.63 | 53,812.05 | 1.17 | 0.254 |
| | Annual production | 2.38 | 0.09 | 25.52 | 0.000 |
| | On the market | −3743.12 | 11,732.38 | −0.32 | 0.752 |
| N | No. of Revenue Streams | 1,173,376.46 | 205,329.81 | 5.71 | 0.000 |

Source: own processing.

Based on results of the ANOVA model with the application of the general equation for multiple regression analysis $Y = \beta_0 + \beta_1 x_1 + \ldots \ldots \beta_j x_i + \varepsilon_i$, we have deduced the predictive relation between examined variables as follows:

$$Y = -12.8 * 10^5 - 4.5 * 10^3 x_1 + 11.9 * 10^5 x_2 + 2.38 x_3 + 11.74 * 10^5 x_4$$

where:

$x_1$ is the cultivated area.

$x_2$ is the number of key business activities.

$x_3$ is the annual production.

$x_4$ is the number of revenue streams.

We can mark this equation as a general form to predict revenue from a cultivated area, the number of key business activities and number of revenue streams is:

By virtue of the unstandardized regression coefficient b1 is equal $-4.5 \times 10^3$ it can be then asserted, that incrementing the value of the independent variable cultivated area by 1 unit (ha) decreases revenue by 4516.51 euros. Increasing the number of key business activities by 1 activity, can also enhance the revenue by $11.9 \times 10^5$ euros. Increasing annual production of wine by 1l can ensure an increase in revenue of 2.38 euros. An increase in number of revenue streams by 1 stream, ensures there is an increase in revenue by $11.74 \times 10^5$ euros.

Table 8 represents the results of the regression model summary and the ANOVA model. The scope of the revenue model (expressed by the number of revenue streams) depends on the following regressors: Number of key business activities, number of off-line channels types, the cultivated area, number of additional business activities and wine tourism activities. The results of the multiple linear regression analysis confirmed the significant influence of all examined predictors. The F-ratio shows whether the overall regression model is a good fit for the data. Outputs indicate that the independent variables statistically significantly predict the dependent variable, $F(5, 97) = 382.97$, $p$–value = 0.000. The model explains up to 80% of the variance of the scope of the revenue model.

**Table 8.** Linear Regression Model Summary and ANOVA results for dependent variable. No. of Revenue Streams.

| Regression | *R* | $R^2$ | *Mean Square* | *F* | *Sig.* |
|---|---|---|---|---|---|
| No. of Revenue Streams | 0.89 | 0.80 | 180.55 | 382.97 | 0.000 |

Source: own processing.

Thereafter, we have looked over the statistical significance of each of the independent variables. As reported in Table 9, not all examined variables and the unstandardized coefficients are equal to zero in the sample: t ($\beta_{Cultivated\ area}$) = 4.86, $p$-value = 0.000; t ($\beta_{No.\ of\ Key\ Business\ activities}$) = 2.11, $p$-value = 0.039; t ($\beta_{Offline\ channels\ types}$) = 0.82, $p$-value = 0.417; t ($\beta_{Additional\ Business\ activities}$) = 13.66, $p$-value = 0.00; t ($\beta_{wine\ tourism\ activities}$) = 2.37, $p$-value = 0.021.

**Table 9.** Summary table for Coefficients. Scope of Revenue Model.

| $X_i$ | Unstandardized Coefficients | | | |
|---|---|---|---|---|
| | **B** | **Std. Error** | **t** | **Sig.** |
| (Constant) | 0.96 | 0.06 | 14.97 | 0.000 |
| No.of Additional Business activities | 0.87 | 0.06 | 13.66 | 0.000 |
| wine tourism activities | 0.17 | 0.07 | 2.37 | 0.021 |
| Cultivated area (ha) | 0.00 | 0.00 | 4.86 | 0.000 |
| No. of Off-line channels types | 0.06 | 0.07 | 0.82 | 0.417 |
| No. of Key Business activities | 0.50 | 0.24 | 2.11 | 0.039 |

Source: own processing.

Based on the results of the ANOVA model with the application of the general equation for multiple regression analysis $Y = \beta_0 + \beta_1 x_1 + \ldots \ldots \beta_j x_i + \varepsilon_i$, we have deduced the predictive relation between examined variables as follows:

$$Y = 0.96 + 0.87x_1 + 0.17x_2 + 0.00x_3 + 0.50x_4$$

where:

$x_1$ is the number of additional business activities.

$x_2$ is the wine tourism activities.

$x_3$ is the cultivated area.

$x_4$ is the number of key business activities.

The intercept is equal to 0.96. Incrementing the value of the independent variable $x_1$ (number of additional business activities) by 1 unit, increases the scope of revenue model by 0.87 unit. An increase in number of wine tourism activities by 1 activity, increases the scope of revenue model by 0.17 units. However, increasing the cultivated area in ha does not cause an increase in the scope of the revenue model. Whereas increasing the number of key business activities by 1 activity, can enhance the scope of the revenue model by 0.5 units.

The text following an equation need not be a new paragraph. Please punctuate equations as regular text.

## 4. Conclusions

In this research paper, we studied the issues of the revenue streams and revenue models researched in the wine-making industry. When creating the theoretical concept of the revenue model, we took into account the premise that company's revenue model can be composed of different revenue streams that can all have different pricing models, different customer value, as well as a different segmentation concept.

The goal of our research article was to invent and design the general revenue model. Concurrently, we have focused on analyzing sources of revenue streams and selected key business variables. This concept was created by using theoretical knowledge about the value, pricing and segmentation. This research goal was fulfilled with two steps.

The reason why we exactly focused on the theme, is due to the fact that according to the author's knowledge, no comprehensive work has been dedicated to the topic of revenue models in companies. Thus, our attempt was to fill in this gap in the business management literature. Our research article belongs to the literature on revenue models and revenue streams, and it brings new information on the influence of economic and financial indicators on revenue and revenue streams in the specific sector. Results of the multiple linear regression analysis confirmed the significant influence of the cultivated area, annual production, years on the market, number of key business activities, number of off-line channel types on our examined variable, "Revenue" and, "Scope of revenue model". These key business activities and the number of off-line channel types are of significant importance to profitability alongside the traditional winery factors such as cultivated area, annual production and years on the market. All of these factors should be added into the overall business model of a winery.

Based on the analyzed data we can state that the revenue model of wineries is a multi-stream revenue model that enables the company to find unique opportunities to generate new revenue and increase the profitability of the company. From the application point of view, without a clearly defined company revenue model, it is impossible and inaccurate to manage and optimize revenues earned from each stream. Likewise, the inadequate construction of the revenue model leads to the risky realization of revenue optimization and can influence revenue leakages. We conclude the paper by suggesting implications for research and practitioners, providing the general concept of revenue model as a tool for managers which allows them to reflect the logic and ways of generating revenues and identify opportunities for revenue optimization and critical issues relevant to price optimization to make it efficient and cost effective.

### Limitations of the Study

In the theoretical background of this research paper, we summarize definitions of the revenue models and revenue streams from well-known foreign authors, who have conducted research analyses in this area for many years. Based on this research, as well as on our own research, we were able to find out and design the general concept of the revenue model, which was applied in wine industry of

Slovakia (depicted in Figure 1). Each revenue stream represents a collection of specified information about the source of revenue and the reason for revenue. Those are afterwards divide into the pricing model and customer segmentation. The pricing model is created as a next step based on price point, value and pricing metrics.

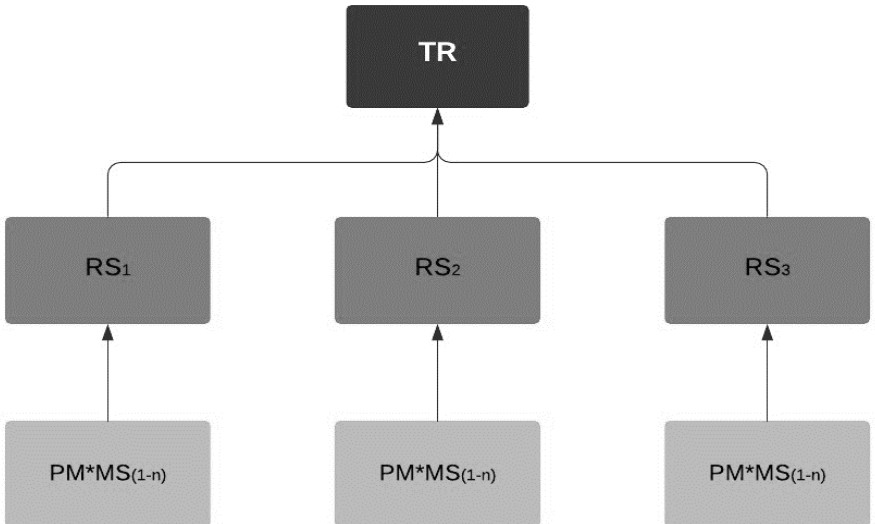

**Figure 1.** General Concept of the Revenue Model©. Source: author's own processing. Where: TR is the total revenue. RS is the revenue stream. PM is the pricing model. MS is the market segment.

Despite these original results, we recognize the research is limited as it is specific to a single country and industry. Since we have tested only companies from the winery industry, the research does not include knowledge about the whole concept of revenue models across industries. Future research should be focused on the granular approach to the design of an effective revenue model including pricing models, pricing metrics and payment systems. It is also necessary to identify customer decision making style, willingness to pay for key and additional business activities of the winery as well as to monitor increased earning potential and better customer value utilization through the pricing models.

**Author Contributions:** K.R. and J.K. designed the study and acquired the data. N.J. and K.R. provided data cleaning and analyzed the data. The draft was prepared by N.J. and J.K. K.R. was responsible for revising the manuscript critically for important intellectual content. All authors have read and agreed to the published version of the manuscript.

**Funding:** This research was supported by the Vega Project 1/0017/20 Changes in the implementation of management functions in the context of the fourth industrial revolution and the adaptation processes of businesses in Slovakia.

**Conflicts of Interest:** The authors declare there is no conflict of interest.

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
