# Peer review of "The General Concept of the Revenue Model for Sustainability Growth"

_sustainability, doi:10.3390/su12166635_

Round 1
Reviewer 1 Report
While you are using wine-making industry as your important source of the research result interpretation, that information could be included in the title of the paper.
That paper will have higher value if Authors include much more fresh papers (published on ir after 2015) that are published in journals indexed in Web of Science and have indexation in Scopus databases.
Look at papers similar to:
Hospitals' Financial Health in Rural and Urban Areas in Poland, Bem, A., (2019) SUSTAINABILITY 11(7)
TOWARDS TREASURY 4.0 THE EVOLVING ROLE OF CORPORATE TREASURY MANAGEMENT FOR 2020 Polak, P. Masquelier, F. MANAGEMENT-JOURNAL OF CONTEMPORARY MANAGEMENT ISSUES 23(2)
Full operating cycle influence on the food and beverages processing firms characteristics, Michalski, G. (2016) AGRICULTURAL ECONOMICS-ZEMEDELSKA EKONOMIKA 62(2)
Determinants of Cost Efficiency: Evidence from Banking Sectors in EU Countries, Belas, J. (2019) ACTA POLYTECHNICA HUNGARICA 16(5)
or similar
Author Response
Dear reviewer,
we are glad to obtain positive feedback on our research article,, The General Concept of the Revenue Model for Sustainability Growth“. We also confirm that we accept all suggestions.
The following recommendations have been incorporated in detail with tracking changes:
,, While you are using the wine-making industry as your important source of the research result interpretation, that information could be included in the title of the paper“.
Answer: The result of the research paper is a new designed General Revenue Model which could be widely used in different business models as a nested model. By this reason, it wouldn’t be correct to narrow the title of the research paper only on the wine-making industry. Wine producing industry had been chosen as an example because there is a significant difference between revenue streams, based on manufacturing (wine-producing) and tourism (hospitality industry) activities. From our point of view, we believe that the specification of the used industry better suits to the abstract of the research paper. We have clarified selected industry in the abstract of the research paper (on the line number 18-19).
- ,,That paper will have higher value if Authors include much more fresh papers (published on ir after 2015) that are published in journals indexed in Web of Science and have indexation in Scopus databases.“ Look at papers similar to:
Hospitals' Financial Health in Rural and Urban Areas in Poland, Bem, A., (2019) SUSTAINABILITY 11(7)
TOWARDS TREASURY 4.0 THE EVOLVING ROLE OF CORPORATE TREASURY MANAGEMENT FOR 2020 Polak, P. Masquelier, F. MANAGEMENT-JOURNAL OF CONTEMPORARY MANAGEMENT ISSUES 23(2)
Full operating cycle influence on the food and beverages processing firms characteristics, Michalski, G. (2016) AGRICULTURAL ECONOMICS-ZEMEDELSKA EKONOMIKA 62(2)
Determinants of Cost Efficiency: Evidence from Banking Sectors in EU Countries, Belas, J. (2019) ACTA POLYTECHNICA HUNGARICA 16(5) or similar
Answer: Thank you for given recommendations on journal articles. We have incorporated those that are in line with our research and the content of the research paper (line numbers: 43, 45, 54, 57, 179)

Reviewer 2 Report
The paper respect the requirements of an academic research article.
It analyses the revenue structure of the wine industry enterprises and proposes a general revenue model.
The abstract should mention that this is a wine producing and related tourism activities sector analysis to avoid misleading the potential reader.
The literature review presents sufficient information for the reader to have a complete picture of the analysed topic.
The data acquisition and research framework is well presented and suited for the studied problem.
The conclusions are supported by the results.
Limitations of the study are presented and well explained.
Author Response
Dear reviewer,
we are glad to obtain positive feedback on our research article,, The General Concept of the Revenue Model for Sustainability Growth“. We also confirm that we accept all suggestions.
The following recommendations have been incorporated in detail with tracking changes:
,,The abstract should mention that this is a wine producing and related tourism activities sector analysis to avoid misleading the potential reader.“
Answer: Wine producing industry had been chosen as an example because there is a significant difference between revenue streams, based on manufacturing (wine-producing) and tourism (hospitality industry) activities. We have clarified selection of the wine-producing industry in the abstract of the research paper on rows 18-19.

Reviewer 3 Report
The research problem was well presented and supported by statistical analyses and the results were verified and validated using appropriate procedures. However, my reservations are raised by the very description of the idea of the model presented in the article. It seems that it is a method of modelling known in econometrics as "soft" or "hierarchical models". I think that the authors should complement this part of the article more accurately and carefully.
Author Response
Dear reviewer,
we are glad to obtain positive feedback on our research article,, The General Concept of the Revenue Model for Sustainability Growth“. We also confirm that we accept all suggestions.
The following recommendations have been incorporated in detail with tracking changes:
- „However, my reservations are raised by the very description of the idea of the model presented in the article. It seems that it is a method of modelling known in econometrics as "soft" or "hierarchical models". I think that the authors should complement this part of the article more accurately and carefully.“
Answer: The result of the research work is a newly designed General Revenue Model, which could be widely used in various business models as a nested model. According to the theory of multilevel modelling, our nested model consists of the following three levels: TR - total revenue, RS - revenue stream and PM - pricing model. The first level represents the scope of revenues from all revenue streams of the company. Revenues are used as a variable. The second level is based on different revenue streams. This section identifies all business activities that should be monetized. The third level is created from pricing models. Each pricing model consists of a price, a price metric, and a market segment. We emphasize that it needs to be adopted the appropriate price for each market segment in the pricing model. Without this, it isn't possible to perform a proper price optimization. At this level of the hierarchical model, the variables are price, metric and segment. The following text was incorporated on line numbers 158-166.

Round 2
Reviewer 3 Report
Overall the papers looks fine after the revision.